# Non-Viral Delivery System and Targeted Bone Disease Therapy

**DOI:** 10.3390/ijms20030565

**Published:** 2019-01-29

**Authors:** Abdul Qadir, Yongguang Gao, Patil Suryaji, Ye Tian, Xiao Lin, Kai Dang, Shanfeng Jiang, Yu Li, Zhiping Miao, Airong Qian

**Affiliations:** 1Lab for Bone Metabolism, Key Lab for Space Biosciences and Biotechnology, School of Life Sciences, Northwestern Polytechnical University, Xi’an 710072, Shaanxi, China; abdulqadirwazir145@yahoo.com (A.Q.); gaoyongguang@nwpu.edu.cn (Y.G.); Suryaji@mail.nwpu.edu.cn (P.S.); tianye@nwpu.edu.cn (Y.T.); linxiao@nwpu.edu.cn (X.L.); dangkai@nwpu.edu.cn (K.D.); jiang2017@nwpu.edu.cn (S.J.); liyu@nwpu.edu.cn (Y.L.); miaozp@nwpu.edu.cn (Z.M.); 2Research Center for Special Medicine and Health Systems Engineering, School of Life Sciences, Northwestern Polytechnical University, Xi’an 710072, Shaanxi, China; 3NPU-UAB Joint Laboratory for Bone Metabolism, School of Life Sciences, Northwestern Polytechnical University, Xi’an 710072, Shaanxi, China

**Keywords:** non-viral delivery system, cationic lipids, bone targeting moieties, bisphosphonates, bone therapeutic agents

## Abstract

Skeletal systems provide support, movement, and protection to the human body. It can be affected by several life suffering bone disorders such as osteoporosis, osteoarthritis, and bone cancers. It is not an easy job to treat bone disorders because of avascular cartilage regions. Treatment with non-specific drug delivery must utilize high doses of systemic administration, which may result in toxicities in non-skeletal tissues and low therapeutic efficacy. Therefore, in order to overcome such limitations, developments in targeted delivery systems are urgently needed. Although the idea of a general targeted delivery system using bone targeting moieties like bisphosphonates, tetracycline, and calcium phosphates emerged a few decades ago, identification of carrier systems like viral and non-viral vectors is a recent approach. Viral vectors have high transfection efficiency but are limited by inducing immunogenicity and oncogenicity. Although non-viral vectors possess low transfection efficiency they are comparatively safe. A number of non-viral vectors including cationic lipids, cationic polymers, and cationic peptides have been developed and used for targeted delivery of DNA, RNA, and drugs to bone tissues or cells with successful consequences. Here we mainly discuss such various non-viral delivery systems with respect to their mechanisms and applications in the specific targeting of bone tissues or cells. Moreover, we discuss possible therapeutic agents that can be delivered against various bone related disorders.

## 1. Introduction

The human body contains soft tissues and organs which are supported and protected by the skeletal system. The skeletal system is composed of bone cells, bone matrix, and bone minerals. Abnormalities in any of these parts may result in bone disorders like bone cancer, osteoarthritis, and osteoporosis [1]. Although systemic administrations can treat skeletal system but are limited due to their high usage of doses that often result in deleterious effects in non-skeletal tissues [2]. Therefore, in order to lower potential levels of toxicity, attain higher drug delivery efficiency, and improve efficacy, targeted therapeutic approaches are needed.

To date, based on biochemical, molecular, and radiologic studies, more than 450 different types of skeletal dysplasia have been characterized [3]. Because of complex anatomical structures and avascular regions, it is very difficult to treat human bones with non-targeted therapies. They are risky because they deliver most of the drugs into visceral organs and have poor biocompatibility, bioavailability, and circulation time [4]. Therefore, at the beginning of the last century, various concepts emerged to treat skeletal disorders with targeted deliveries.

Currently, a number of targeted delivery systems against bone disorders are available or are under consideration. Drugs can be delivered either by targeting the whole skeletal system or specific cellular locations in it. The pathogenesis of some diseases, including metabolic skeletal dysplasia, have been treated by targeting the entire skeletal system. However, specific targeting involves the delivery to the bone resorption cells, i.e., osteoclasts or osteoblasts; or bone formation cells, which is the focus of modern research [5]. In order to develop effective drug delivery, it is always important to select the most reliable targeting moiety. In the last few years a number of targeting moieties such as tetracycline and bisphosphonates (BPs) have been synthesized. Bone marrow cells have also been discovered to be used as a targeting moiety [6]. Moreover, nanoparticles like lipid nanoparticles (LNPs), liposomes, and Poly (lactic-co-glycolic acid) (PLGA) nanoparticles are also used as drug carriers. The US Food and Drug Administration (FDA) has also approved some nanoparticles which can be used as drug carriers in the treatment of skeletal disorders [7]. Apart from this, some viral vectors like lentiviral vectors, adenovirus vectors, adeno-associated virus vectors (AAVs) and retroviral vectors are also used as bone-targeting RNA delivery systems.

Living cells do not receive naked DNA, RNA, and certain drugs deliberately because of their large sizes and hydrophilicity problems. Moreover, enzymes like nucleases may degrade naked DNA or RNA [8]. Therefore, it is necessary to construct the vectors that can carry and deliver the required DNA, RNA or drugs specifically to the target cells. Generally, two types of delivery systems are used to deliver into the living system; the one using viral vectors are called viral delivery systems, while the others using non-viral vectors are called non-viral delivery systems. Viral delivery systems possess high-transfection efficiency but being oncogenic, immunogenic, and capable of transporting only small size DNA molecules they are limited. While, non-viral vectors are comparatively safe, having low immunogenicity, easy to manufacture, and are flexible in targeting to lower off-target toxicity. However, a major problem with non-viral systems is their low transfection efficiency, colloidal instability, and clustering in circulation [9]. But efforts are still in progress to minimize such drawbacks. Over the last decade, a number of non-viral vectors have been manufactured, improved, and used for the purpose of gene, siRNA, and drug delivery. Apart from these systems, physical methods including needle injection, balistic DNA injection, electroporation, sonoporation, photoporation, magnetofection, and hydroporation are also used for transfection.

In this review, we mainly discuss about the different kinds of non-viral delivery systems, their mechanistic features, and applications in the form of bone targeting moieties in the treatment of bone-related disorders at the molecular level.

## 2. Non-Viral Delivery Systems

Non-viral delivery system simply refers to the use of non-viral vectors for drugs, genes or RNA delivery into cells. Non-viral vectors are safer, less toxic non-immunogenic alternatives to the viral vectors. They are easy to design, develop, and are inexpensive. Many of the non-viral vectors have been extensively used for the treatment of bones, ranging from bone repair and regeneration to various serious bone disorders like bone cancer, osteoarthritis, and osteoporosis (Table 1).

Non-viral vectors are broadly classified as organic and inorganic vectors. Organic non-viral vectors include cationic lipids, polymers, and peptide-based vectors while those that are inorganic in nature are calcium phosphates, calcium carbonate, and metal nanoparticles. Hence, non-viral delivery systems can be categorized on the basis of the types of vectors they use into organic and inorganic delivery systems. In order to understand them in detail, they are discussed both in general and as well as with some specificities to bone as below.

### 2.1. Organic Delivery System

#### 2.1.1. Cationic Lipids

Since the middle 1980s, synthetically-prepared, positively-charged cationic lipids were considered effective delivery vehicles for proteins and nucleic acids. Before this, they were used for purposes like membrane fusion mechanisms and protein studies [10,11]. Felgner and his coworkers in 1987 were the first to use cationic lipid (DOTMA) as a gene transfer vehicle [12]. Cationic lipids such as DOTMA (N-[1-(2,3-dioleyloxy)propyl]-N,N,N-trimethylammonium chloride), DOTAP(1,2-dioleoyl-3-trimethylammoniumpropane), DDAB (Didodecyldimethylammonium bromide), CTAB (Cetyltrimethylammonium bromide) (Figure 1) are considered to be the most appointed and well-studied non-viral vectors. Being positively charged they bond and self-assemble easily with negatively-charged nucleic acids making complexes called lipoplexes. They also bind to negatively charged cell membranes more readily than conventional liposomes [13].

Over the last ten years, a large number of cationic lipids have been developed and applied for the purpose of drug delivery especially nucleic acids. Some of them like lipofectamine and DOTAP are commercially available and frequently used as the gold standards in different cellular experiments for sufficient transfection efficiency [14]. They all have positively charged hydrophilic heads and hydrophobic tails, connected to each other with a linker. The hydrophilic head groups are mostly primary, secondary, tertiary amines, or quaternary ammonium salts, but some other groups like imidazole, guanidino, phosphorus, pyridinium, and arsenic groups have also been established. While the hydrophobic tails contain aliphatic chains, cholesterol, other types of steroid rings or rigid aromatic rings. Usually the linkage in between the head and tail is made by ester, ether carbamate, or amide groups [8]. This is the positively charged head that bind with negatively charged nucleic acids and linker that affect the rate of biodegradability.

Cationic lipids are comparatively less immunogenic, less toxic and easily producible. However, their transfection efficiency needs improvement. Moreover, they form aggregates in the blood causing toxicity and inflammatory response. But these drawbacks are treatable by designing ionizable cationic lipids with PEGlyated lipids in formulation. PEGlyation prevent the cationic lipids from opsonization by reticulo-endothelial system, and thus give the opportunity for prolonging in vivo circulation [15].

Cationic lipids have been used as successful delivery systems to different types of cells, including bone cells, endothelial cells, airway epithelial cells, muscle cells, placental cells, hepatocytes, tumors, and others. Recently, Attia and her colleagues developed a delivery system in the form of niosomes by using cationic lipid called 2,3-di(tetradecyloxy)propan-1-amine. They used these niosomes to deliver the *BMP-7* gene to mesenchymal stem cells (MSCs). The *BMP-7* gene actually codes for bone morphogenetic protein-7(BMP-7) that plays an important role in transforming mesenchymal stem cells (MSCs) into bone. An enhanced growth rate with extracellular matrix deposition and promoted alkaline phosphatase activity (ALP) was observed in transfected MSCs, thus suggesting the formation of osteoblasts-like cells. They concluded that their designed delivery tool can be used not only as efficient delivery system for *BMP-7* but also as proliferating and bone forming cells for bone regeneration [16]. Xuelei Yin and his colleagues developed estrogen-functionalized liposomes grafted with glutathione-responsive sheddable chotooligosaccharides against osteosarcoma. They found that Chol-SS-COS/ES/DOX liposomes manifested higher cytotoxicity to MG63 osteosarcoma cells than to liver cells [17]. Moreover, our group recently designed a delivery system by deriving cationic lipids from [12]-aneN_3_ and modified it with fluorescent naphthalimide, oleic acid and octadecylamine. We found that all of them showed good transfection efficiency to osteoblastic cell line MC3T3-E1, MG63, HeLa, and HEK293 cells, but the one modified with naphthalimide showed even higher efficiency than lipofectamine 2000. Most importantly, it was successfully applied for in-situ monitoring of cellular uptake, DNA transportation, and release through non-invasive fluorescence imaging. Hence, we concluded that it can be used as a multifunctional non-viral delivery system for treating various bone disorders related to osteoblasts in future [18].

#### 2.1.2. Cationic Polymers

Polymeric systems containing positive charges are known as cationic polymers. Being positively charged, they can bind with negatively-charged nucleic acids, proteins, and cell membranes through electrostatic interaction. When they are mixed with DNA they form complexes called polyplexes, usually more stable than lipoplexes [19]. They are considered to be excellent nucleic acids transfer vectors as they mediate the transfection through condensation of nucleic acids, facilitate their uptake by cells, protect them from nucleases, and help in endolysosomal escape. Moreover, they have been developed for use in other applications like drug delivery and tissue engineering.

In 1995, Boussif and his colleagues [20] were the first to use a cationic polymer called polyethylenimine (PEI) as a gene delivery vehicle. Now a great variety of cationic polymers have been synthesized and analyzed for their gene transfer ability. Cationic polymers may be either natural or synthetically developed. Natural cationic polymers include chitosan, cationic dextran, gelatin, cationic cellulose, and cationic cyclodextrin, while polyethylenimine (PEI), polyamidoamine (PAA), polyaminoester (PAE), poly-*L*-lysine (PLL), and poly dimethylaminoethylmethacrylate (PDMAEMA) (Figure 2) are the well-known synthetic cationic polymers [21]. Natural cationic polymers are generally non-toxic, less immunogenic, and biodegradable. Therefore, many of them like gelatin and cationic dextran have been approved by the FDA as safe biomaterials [22]. Synthetic polymers have the advantage of modification according to the situation by incorporating bioactive moieties and functional groups.

Polyethylenimine (PEI) is the most important and highly used cationic polymer, synthesized in both linear and branched form with various molecular weights. It consists of reactive amino groups, and thus can be modified chemically for desirable physiochemical properties. The positively charged amino group of PEI can bind with negatively charged molecules including drugs and nucleic acids. However, they are somewhat cytotoxic and non-degradable, but they have the ability to form small and enzymatically stable polyplexes leading to high transfection efficiency [23].

Generally, cationic polymers with higher molecular weights and positive charges can act as efficient delivery systems but unfortunately they are limited due to their excessive cytotoxic effects [24]. Moreover, being highly positively charged, they can bind with anionic biomolecules in serum and thus may result in off-target side effects [25]. On the other hand, cationic polymers with low molecular weights possess nice cell tolerability but it is a pity that they are less efficient as compared to their higher molecular weight counterparts [26]. In order to break such a paradox between cytotoxicity and efficiency, recently, Liu and his fellows synthesized low molecular weight cationic polymers (LCPs) modified with coordinated zinc. They reported that Zn coordinated ligand bind with high affinity to the phosphate group in DNA that help the formulated polyplex in endosomal escape and higher transfection efficacy [27]. Recently, Bilecen and his colleagues developed a novel delivery system against osteoporosis by making a complex between PEI and RANK (Receptor activator of nuclear factor κ B) siRNA, and loaded it into poly(lactic acid-co-glycolic acid) (PLGA) nano-capsules. A reduction of about (47%) RANK mRNA levels was observed on osteoclast precursors which is directly proportional to the suppression of differentiation to mature osteoclasts. Hence they declared that this delivery system may reduce the number of mature osteoclasts that leads to the treatment of osteoporosis [28].

#### 2.1.3. Cationic Peptides

Various cationic peptides are under investigation for use as safe and sound vehicles for targeted nucleic acids and negatively charged drug delivery. As they are rich in positively-charged lysine and arginine residues, therefore, they can bind with DNA efficiently and condense it into small compact particles. They are comparatively more advantageous because they have the ability to interact electrostatically with negatively charged nucleic acid or drugs, protect and target it to specific cell receptors, disrupt endosomal membrane, and deliver the cargo into nuclear localization [29]. Short peptides are readily attached to lipoplexes and polyplexes in order to give them directions for achieving specific targets in cell [8].

Recently, Mansure and his colleagues synthesized novel polymer–drug–peptide conjugates containing carboxymethylcellulose (CMC) conjugated by amide bonds with an anticancer drug called doxorubicin. In order to facilitate the targeting and internalization by cancer cells they dually functionalized the conjugate with L-arginine (R) and integrin-target receptor tripeptide (RGD). They tested and observed the bioconjugates against breast, bone, and brain cancer cell lines and reported them as ‘‘smart’’ delivery system to normal cells while highly toxic to cancer cells [30]. Sun and colleagues reacted cationic peptide SDSSD (Ser, Asp, Ser, Ser, Asp) with polyurethane (PU) nanomicells and developed a tremendous osteoblast-targeting delivery system, i.e., SDSSD-(PU) that specifically binds to osteoblasts through a surface protein called periostin. They utilized this system for the delivery of siRNA/microRNA to osteoblast and got successful results. They found out that SDSSD-PU could specifically target osteoblasts both in vitro and in vivo; therefore, it can be used as an effective osteoblast-targeting small nucleic acid delivery systems to treat skeletal disorders [31].

### 2.2. Inorganic Delivery System

#### 2.2.1. Calcium Phosphates

In 1973, Graham and Van der Eb were the first to use calcium phosphates as non-viral vectors [32]. Now they are considered among the most prominent of non-viral delivery systems. They contain divalent cations, i.e., Ca^2+^ that form stable ionic complexes with negatively charged nucleic acids. These complexes are transferred to the cells through ion channel-mediated endocytosis. Although they possess low transfection efficiency, they are highly compatible and easily biodegradable. They are comparatively less toxic than commercial transfection reagents like Lipofectamine 2000, therefore they are considered as safe materials for transfection [33]. Moreover, they overwhelm the other targeting problems including endosomal escape, protection of the cargo in cytosol, and its delivery into the nucleus.

Calcium phosphate nanoparticles can be used in the treatment of various bone disorders by carrying drugs like bisphosphonates and other related drugs to the target sites [34]. Recently, Kentaro and his colleagues tested calcium phosphate cement (CPC) as a releasing and delivery material for a vancomycin antibiotic comparative to that of standard polymethylmethacrylate (PMMA) cement. They reported that calcium phosphate cement released more vancomycin both in vivo and in vitro for a longer time than PMMA, hence it can be used as an appropriate material for releasing antibiotics against postoperative infection [35].

#### 2.2.2. Metal Nanoparticles

In the new era, nanotechnology has gained a lot of attention for developing new methods regarding diagnosis and treatment of diseases [36]. Various nanomaterials including grapheme, liposome, carbon nanotubes, magnetic nanoparticles, and metal nanoparticles have been utilized in many biomedical therapies. As they are nanosized, therefore it is easy to manipulate them at molecular level. Moreover, they possess large surface area, high compatibility, and strong antioxidant property [37].

Metal nanoparticles especially gold nanoparticles (Au NPs) are of prime importance for use in various biomedical applications as they can be designed in different sizes and different shapes. They are non-toxic and highly biocompatible. Moreover, they are electrostatically charged and hence can be functionalized by other biomolecules like nucleic acids and drugs [38]. They can be used to transfer multiple drugs molecules, nucleic acids or vaccines into the target sites with controlled release. They are able to make direct conjugates with different drugs molecules through ionic, covalent or physical absorption to treat endocellular diseases [39]. Lee and his colleagues made a conjugate of gold nanoparticle with alendronate of the bisphosphonate group and checked their inhibitory effects on the receptor activator of nuclear factor-κb ligand (RANKL)-induced osteoclastogenesis in bone marrow-derived macrophages. They observed that GNPs-ALD (gold nanoparticles conjugated with Alendronate) had suppressed osteoclast formation, thus declaring them as efficient agents for treating osteoporosis [40]. Apart from gold nanoparticles, iron oxide (Fe3O4) nanoparticles, which are safe, cheap, and chemically stable, can also be used to trigger osteoclast regulation. Actually, they possess high magnetic fields which are responsible for increasing local temperature, and in this way, suppress osteoclast functions to treat osteoporosis [41].

## 3. Mechanism of Non-Viral Delivery to Bone Cells

Non-viral delivery systems, unlike viruses, do not have the ability of inserting genes directly into the cells’ nuclei. In order to reach and enter the cells, they always face some extracellular and intracellular obstacles both in vitro and in vivo. To date, complete mechanisms of non-viral delivery towards bone cells is still skeptical. However, according to the available literature, generally it can be divided into four stages including binding to cell surface, endocytosis and intracellular processing, endolysosomal escape, and nuclear entry and expression. As different types of non-viral vectors may follow slightly different mechanisms, therefore, for the sake of easiness, we will refer to the most extensively used non-viral vectors, i.e., cationic lipids and cationic polymers in terms of DNA delivery to bone cells and try to understand the mechanisms. 

### 3.1. Binding to Cell Surface

Cationic lipids and cationic polymers are positively charged; therefore, they have the ability to bind with negatively charged DNA and cell surfaces through electrostatic interactions. Upon mixing with DNA, cationic lipids form complexes called lipoplexes while cationic polymers form polyplexes. Usually, complexes are assembled with a minimal surplus positive charge to allow them to bind with the negatively charged cell surfaces. High charge ratios lead to the formation of small complexes while larger aggregates are formed when the net charge is near to zero [42].

Generally, cell surfaces are negatively charged because of the presence of proteoglycans with different glycosaminoglycan chains including dermatan, heparin, and chondroitin sulfates. Especially heparan sulfate proteoglycans (HSPGs) are assumed as the most prominent interacting regions for cationic delivery vectors [43]. Moreover, they are also used by some viruses and bacteria to get entry into the cells [44]. They contain small actin-rich cellular extensions called filopodia that act as antennae for probing the extracellular environment [45]. It has been shown that interaction of lipo-polyplexes with cell surfaces result in rapid clustering of single transmembrane domain proteins called syndecans, which is a major class of HSPGs that stabilize the binding of nanoparticles for retraction towards the cell surface through filopodia [46].

In general, polyplexes face degradative enzymes and negatively charged serum proteins in vivo. If cationic polymers fail to condense and compact, the DNA properly degradative enzymes will digest it [47]. Similarly, negatively charged serum proteins may bind to them and disturb the transfection efficiency. However, this issue can be solved by binding complexes with hydrophilic polymers like polyethylene glycol (PEG) [48].

### 3.2. Endocytosis and Intracellular Processing 

Endocytosis is the process through which biomolecules and other cargo enter into the cells. Generally, it can be divided into two categories, i.e., phagocytosis and pinocytosis. Phagocytosis refers to engulfing large solid particles while pinocytosis for taking up solutes, fluid, and smaller particles. Unlike phagocytosis, pinocytosis generally occurs in all types of cells, and therefore its related pathways are of prime importance in delivery. The most prominent pinocytosis pathways include clathrin-mediated endocytosis, caveolin-mediated endocytosis, clathrin- and caveolin-independent endocytosis and macropinocytosis [49] (Figure 3). All of these pathways engulf the biomolecules and other cargos like non-viral vectors, and thus form various sizes of vesicles that eventually integrate into endosome. However, each pathway may deal the internalized vectors in different ways [50].

Clathrin-dependent endocytosis pathway exploits receptors including transferrin receptors and clathrin-coated pits. Polymeric vectors first bind to these receptors, are taken up, assemble in vesicles, and are moved to the early endosome for facing acidified environment, early endosome to late endosome, and finally degraded in lysosomes [51]. Whereas, caveolin-dependent endocytosis associates with receptors like folate receptors and lipid rafts, and thus ignore the acidic environment and lysosomal degradation [52].

### 3.3. Endolysosomal Escape

Endolysosomal escape is considered to be the one of the most important factors that imitate the delivery efficiency of non-viral vectors. The internalized molecules after internalization through endocytosis are situated in endosomes. They should be released into cytoplasm otherwise they will be destroyed by the acidic environment of endosomes. Moreover, endosomes may fuse with lysosomes and hence degraded [53]. Therefore, for efficient delivery they must get release of endosome through a process called endosomal escape. 

In order to get successful endosomal escape, different strategies have been adopted for different types of complexes. Some lipoplexes consist of DOPE (dioleoyl phosphatidylethanolamine) which is basically a pH-sensitive fusiogenic lipid that form stable lipid bilayers at neutral pH but transits to inverted hexagonal structures at acidic pH. Because of the acidic environment inside endosomes, DOPE assumes an inverted hexagonal structure that tightly binds and disturbs the stability of endosomal membrane, and thus releases the contents into the cytosol [54]. Similarly, PEI has the same intrinsic ability of causing endosomal release but with different mechanism. Behr and his coworkers suggested that PEI is more protonated in acidic pH of endososme. It causes the influx of chlorine ions with water leading to swelling and finally rupturing of endosome to release the complexes. This hypothesis is called proton sponge hypothesis [55]. Moreover, use of lysosomotropic reagents such as chloroquine is also a reliable source for causing endosomal escape. It is basically a weak hydrophobic base that is protonated in the acidic environment of lysosome. This protonation causes swelling and disturbs their cell membranes stability [56].

### 3.4. Nuclear Entry and Expression 

After successful endolysosomal escape, the lipo and polyplexes should move through cytosol towards nuclear membrane. Although some naked DNAs can pass through nuclear membrane by diffusion, it is difficult for the larger DNA molecules to do so. Moreover, nucleases which are situated in cytoplasm may degrade the DNA and thus reduce the transfection efficiency [57]. Therefore, passive transport is not enough to move the polyplexes towards nuclear membrane [58]. Fluorescent particle tracking experiments have revealed that microtubules and filaments actively transport polyplexes into the nucleus using molecular motors [59]. Therefore, breakage of microfilaments and microtubules resulted in 60–80% and 75% reduction in PEI gene delivery respectively [60].

Generally, two types of pathways, i.e., cell-dividing dependent and cell-dividing-independent pathways, are considered to be responsible for nuclear entry of DNAs alone or in the form of non-viral vectors [61]. Large molecules especially polyplexes enter the nucleus through cell-dividing dependent pathways. They can get their entry only when the nuclear envelope is broken down and the dividing cell is in a G2/M transition state [62]. Hence, this pathway is only reliable for dividing cells. On the other hand, in cell-dividing-independent pathways, small molecules like lipoplexes have to pass through double-layered nuclear membranes either through nuclear pores or active transport using kariophilic proteins as transfer agents [63]. This pathway is specialized for non-dividing cells. Nuclear membranes may diffuse cargo molecules of about 9 nm in diameter, while the active transport needs nuclear localization signals to interact the cargo with nuclear import machinery for crossing the nuclear pore complex [64]. Nuclear localization signals, like the typical nuclear location sequence taken from SV40 large T-antigen and the bipartite nuclear location sequence, are attached with cargo proteins to interact with importin of nuclear pore. This interaction facilitates the complex in crossing the nuclear membrane successfully [65]. The DNA is released from complexes due to the electrostatic interaction of highly negatively-charged chromatin proteins with positively-charged cationic lipids or polymers, and thus exchanges it with cellular DNA [66]. The delivered DNA contains promoter sequences where transcription factors interact and allow the RNA polymerases to make RNA using that DNA as a template. The mRNA is carried out to the cytosol to translate it into the desired protein as usual. Here, non-viral gene delivery system research is somewhat inappropriate as sometimes high transfection rates results in very low or undetectable expressions [67].

## 4. Bone Targeting Systems

In order to avoid the risks of toxicity to non-targeted tissues and low therapeutic efficiency bones are targeted with a specific delivery system. Targeting moiety works as an important part in targeted delivery system as they help in carrying the cargo towards its target. They are categorized as bone specific and bone cell specific moieties. The bone specific moieties target the whole skeleton (Figure 4) while bone cell specific moieties specifically target osteoclasts and osteoblasts cells (Figure 5). Targeting whole the skeleton may have deleterious side effects in the form of distribution to non-targeted cells. However, cell specific delivery systems concern specific cells and do not disturb or harm the normal cells. Therefore, they are considered as the more suitable and desirable delivery system.

### 4.1. Bone Specific Targeting Moieties

#### 4.1.1. Bisphosphonates

In 1969, Fleisch and his colleagues [68] for the first time published bisphosphonates as inhibitory materials for bone resorption and hydroxyapatite dissolution. A few years later they were described as preventive agents against osteoporosis in rats [69]. Actually, they constitute a family of synthetic drugs extensively used in clinics against diseases causing loss of bones densities like osteoporosis, bone metastasis, osteogenesis imperfecta, etc. They are called bisphosphonates because they contain central carbon bonded to two terminal phosphate groups (P–C–P); therefore, they are also called as diphosphonates. R-group (Alkyl group) or side chains of central carbon are easily approachable and can be manipulated according to the situation. Their mechanism is well defined as they are negatively charged and thus interact electrostatically with positively charged hydroxyapatite (HA) of bones after administration [70]. Their affinities to hydroxyapatite differ because of varying side chains [71]. They have the ability of inhibiting bone resorption by osteoclasts and promoting bone formation by osteoblast differentiation simultaneously [72]. They have been used as inhibitors against bone turnover in Paget’s disease, malignancy of the bone, and osteoporosis [73]. However, they have some drawbacks in the form of low bioavailability and deleterious side effects like osteonecrosis of jaw, musculoskeletal pain, and ulcers [74].

Due to safety measures the optimal administration of bisphosphonate is under investigation. Its prolonged usage may have negative effects on bone mass. Because of long term usage of BPs, some of the patients suffering from osteoporosis have been detected with disorders of osteonecrosis of the jaw and atypical femoral fracture [75]. In one of the study, patients of osteoporosis affected with osteonecrosis of the jaw have been estimated 1 in 10,000 [76]. Some cancers like breast cancer and prostate cancer usually metastasize to bone. Their treatment with BPs can delay their time to metastasize to bone and reduce tumor growth. Multiple agencies have approved zoledronate as preventive material from the onset of skeletal related events in non-small cell lung cancer and thus palliate pain [77]. However, it has been reported that, 95% of the cancer patients have developed jaw osteonecrosis after intravenous reception of high doses of BPs [78]. Moreover, BPs have been used against primary bone tumors along with osteosarcoma to reduce bone lesions and pain [79]. Cheng and his colleagues [80] found reduced MMP-2 and invasion in human osteosarcoma cell lines when they were treated with alendronate. 

Apart from systemic administration, calcium phosphate is considered to be an appropriate biomaterial for BP’s delivery to bone. Calcium phosphates are the main parts of bones; therefore, they are comparatively more compatible and degradable. Combination of calcium phosphate with bisphosphonates like alendronate may act as a local implant drug for promoting bone formation in an osteoporotic model [81]. Faucheux and colleagues [82] performed an in vitro experiment and reported that zoledronate can be loaded by calcium phosphate biomaterial and released in a controlled manner to inhibit osteoblastic resorption without disturbing osteoblast.

Nanoparticles usage for BP targeting is another approach for drug delivery to bone. BPs targeting moieties can be attached to nanoparticles to carry it to the bone sites. Nanoparticles have the advantage of carrying huge amounts of a drug, protecting it from degradation, and extending its circulation time to target the particles to bone [83]. New technologies and polymer design have increased the graph of encasing drugs in nanoparticles and deliver them to specific targets. Many conjugates of BP nanoparticles have been reported with improved drug efficiency and efficacy in pre-clinical studies. In one of the study, PLGA nanoparticles coated with zolendronate were declared as bone targeting materials [84]. A conjugate of PLGA with alendronate encapsulating doxorubicin resulted in more reduction in bone metastases than the free drug delivery [85]. Like polymers, liposomes can also make conjugates with BPs to target specific sites in bone. Adriamycin encased in liposome with BP head groups was considered more effective and toxic to osteosarcoma cells in vitro than the drug alone or without BP head groups [86].

#### 4.1.2. Tetracycline

Tetracycline constitutes a family of broad-spectrum antibiotics that were first discovered in the 1940s. They are isolated either directly from the *Streptomyces* genus of bacteria or developed synthetically using isolated compounds from it. Traditionally they are exploited against many bacterial infections but they can also be utilized for bone targeting because of their binding affinity with hydroxyapatite of bone [87]. Wang and his colleagues developed nanoparticles by making conjugates of tetracycline with PLGA. They reported that these nanoparticles have the ability to target bone and transport hydrophobic drugs like simvastatin to treat osteoporosis [88]. Recently, Gomes and his fellows demonstrated that doxycyclines decrease osteoclasts, increase osteoblast, activate Wnt-1b, and neutralize Dkk-1, and hence may act as a potent material for bone repairing in periodontal diseases [89]. Moreover, tetracyclines are comparatively safer to BPs and do not cause osteonecrosis of jaw and other related disorders.

#### 4.1.3. Oligopeptides

To date, many oligopeptide conjugated drugs have been utilized against several diseases like osteoporosis, musculoskeletal diseases, infection diseases, and cancers. In contrast to polypeptides, oligopeptides contain a small number of amino acids (maximum 10–50). Nowadays, oligopeptides are considered among the potent classes of molecules for nanotechnology applications. Oligopeptides have been reported as materials having strong binding affinity to hydroxyapetite which is the main component of bone [90]. Park and his fellows designed a cyclized oligopeptide against DKK1-low density lipoprotein receptor-related protein (LRP) 5/6 interaction and found reduced tumor burden in results after treatment with it, as a high level of DKK1 often results in osteolytic bone lesions in multiple myeloma models [91]. Hence oligopeptides can act as successful targeting moieties to bone. However, one of the major drawbacks of oligopeptide-based drugs is the enhanced proteolytic instability compared to not only tiny molecules but also monoclonal antibody therapeutics.

### 4.2. Bone Cells Specific Targeting Moieties

#### 4.2.1. Osteoblasts Targeting

Osteoblasts, also called as bone forming cells are most commonly found on outer surfaces of bones. They are always kept in balance to osteoclasts through homeostasis to maintain total bone mass. Any imbalance in bone formation by osteoblasts or resorption by osteoclasts can lead to severe bone disorders [34]. Usually some drugs like parathyroid hormones are used to enhance bone formation but they are limited due to their side effects [92]. Therefore, in order to overcome such limitations development in osteoblasts targeting drugs is required. 

Gene therapy is a new strategy that can be used to deliver small exogenous nucleic acid molecules like miRNAs and siRNAs to combat various diseases [93]. Vectors used for delivery may be either viral or non-viral in nature. However non-viral vectors are comparatively safe and low immunogenic; therefore, they are preferred for osteoblast targeting delivery. Some peptides and aptamers have been developed that can be used as promising osteoblast targeting moieties. In an experimental model a peptide SDSSD was used to develop an osteoblast-targeting delivery system, i.e., SDSSD-PU, where SDSSD binds specifically to osteoblasts through periostin and PU are nanomicells that act as a carrier for siRNA/microRNA. After conducting in vitro and in vivo experiments, the system was concluded as an efficient osteoblast targeting delivery system [31]. Liang and his coworkers used cell-SELEX (systematic evolution of ligands by exponential enrichment) and screened an aptamer, i.e., CH6 that specifically targets both human and rat osteoblasts. They utilized this aptamer to target lipid nanoparticles containing an siRNA against Plekho1gene. Promotion in bone formation was noted because of the silencing of the *Plekho1* gene by successfully delivered siRNA to osteoblasts [94]. Therefore, aptamer functionalized LNPs (lipid nonviral particles) can act as a new RNAi-based strategy for osteoblast targeting. In another experimental model, Zhang and his colleagues designed cationic liposomes based on dioleoyl trimethylammonium propane (DOTAP) and bound six repetitive sequences of aspartate, serine, serine ((AspSerSer)_6_). They used these liposomes as delivery systems for siRNAs to target casein kinase-2 interacting proteins-1 which are encoded by *Plekho1* genes. They found successfully targeted siRNAs delivery against the target bone inhibitory *Plekho1* genes by observing increased bone mass both in normal and osteoporotic mouse models [95].

#### 4.2.2. Osteoclasts Targeting

Osteoclasts, also called bone resorption cells because they digest bone tissues by secreting acids and proteolytic enzymes during healing and growth. They are required in proper balance with osteoblasts to maintain adequate bone mass, otherwise it may lead to serious disorders like osteoporosis, bone tumors, and Paget’s disease [96]. Therefore, it is important to develop an osteoclast targeting delivery system for osteoclast targeting drugs. Mizoguchi and his colleagues reported that miR-31 can act as an important miRNA for osteoclast regulation. They induced the expression of miR-31 with RANKL which controlled the formation of osteoclasts and hence bone resorption by regulating the cytoskeleton organization [97]. In a study, a site-specific bone-targeting drug-delivery system was developed on the basis of *N*-(2-hydroxypropyl)methacrylamide (HPMA) copolymers with d-aspartic acid octapeptide (d-Asp8) having siRNA against *sema4D*. Successful uptake and intracellular trafficking was found within osteoclasts that prevented the suppression of osteoblast activity [98]. Recently, Zhao and coworkers found that miR-340 can stop the differentiation of osteoclasts by suppressing *MITF* (microphthalmia-associated transcription factor) expression. It gives a clue that they can be utilized as most promising therapeutic targets to treat the disorders related to osteoclasts [99].

#### 4.2.3. Targeting with Bone Marrow Mesenchymal Stromal Cells

Bone marrow mesenchymal stromal cells (BMSCs) are non-hematopoietic in origin having the ability to differentiate into different cell types that is, adipocytes chondrocytes and osteoblasts [100]. Therefore, targeting with BMSCs can help in the repair and regeneration of bone tissues. In a study, BMSCs were transfected with liposomal vector containing a human gene cloned from hepatocytes. When they were co-cultured with Raju cells they showed inhibitory effects on tumor growth through the process of lymphoma cell apoptosis [101]. Li and colleagues reported that BMSCs can be used as a target to treat age-related bone loss by regulating miR-188 in them [102].

### 4.3. Therapeutic Agents for Bone-Related Diseases

Some therapeutic agents have been discovered that target specific molecules expressed in bone cells (Table 2). By doing so they either inhibit or promote bones cell functions, hence treating different bone-related disease like osteoporosis and osteoarthritis in return.

#### 4.3.1. Cathepsin K Inhibitors

Cathepsin K (Cat K), is basically a cystein protease member from cathepsin lysosomal protease family which is highly expressed in osteoclasts [103]. It has the ability to destabilize the bone matrix by degrading its main components like hydroxyapetite and type 1 collagen protein, hence, helpful in bone resorption [104]. Therefore, developing Cat K inhibitors against Cat K expression is therapeutically important for treating disorders with excessive bone loss like osteoporosis [105]. To date, several Cat K inhibitors have been developed either from natural products or synthetic compounds and are under investigation at different stages of clinical or preclinical trials. Recently, Lindstrom and his colleagues found that a cathepsin K inhibitor MIV-711 can reduce biomarkers of bone resorption in experimental models of osteoarthritis. Therefore, they suggested that it can be used as a potent disease modifying agent with similar consequences in human patients with osteoarthritis [106].

#### 4.3.2. V-ATPase Inhibitor

The vacuolar ATPase (V-ATPase) is actually a multi-subunit enzyme that exploits ATP hydrolysis to pump the protons across a membrane. It is predominantly expressed in osteoclasts and plays a key role in regulating their extracellular acidity and bone resorption [107]. Therefore, it is clinically important to target it with different inhibitors, i.e., V-ATPase inhibitors to treat disorders like osteoporosis and bone cancers. Tremendous efforts have been made to design and develop selective and potent V-ATPase inhibitors. Crasto and his colleagues [108] reported that luteolin can reduces bone resorption by disrupting interaction between a3 and d2 subunits of V-ATPases, hence acting as an effective V-ATPase inhibitor.

#### 4.3.3. α_v_β_3_ Integrin Receptor Antagonist

Integrins constitute a family of cell surface receptors consisting of an α and β subunit, each having a short cytoplasmic and large extracellular domain. Actually they are heterodimeric glycoproteins responsible for mediating cell–cell and cell–matrix interactions [109]. It has been shown that apart from other types of integrins, α_v_β_3_ integrins are expressed highly in osteoclasts where they act as receptors for various extracellular matrix proteins like osteopontin, vitronectin, and bone sialoprotein [110]. They have been proved as active players in regulating osteoclasts by inhibiting bone resorption through binding with Arg-Gly-Asp (RGD)mimetics, Arg-Gly-Asp (RGD)-containing peptides and blocking antibodies [109,111,112]. Recently Yuval and colleagues developed a bispecific protein antagonist that simultaneously binds to c-FMS (colony-stimulating factor 1 receptor) and α_v_β_3_ integrins receptors. They reported that this protein inhibited bone resorption by disturbing osteoclast activity in a mouse model with osteoporosis, hence it may act as a model therapeutic agent for bone disorders [113]. 

#### 4.3.4. Src SH2 Inhibitors

*Src* is basically a gene that encodes an enzyme nonreceptor tyrosine kinase that phosphorylates other proteins on specific tyrosine residues. It consists of several domains including SH2 domain, SH3 domain, catalytic kinase domain, N-terminal domain, and C-terminal domain [114]. They express in osteoclasts and play an important role in bone resorption. Moreover, SH2 domain of Src is considered to be the main regulator of osteoclast activities [115]. Therefore, Src can act as suitable therapeutic target for Src SH2 inhibitors to inhibit bone resorption. De Vries and colleagues tested the effect of an Src SH2 inhibitor, for example AZD0530, on osteoclasts. They reported that AZD0530 reversibly affected osteoclast formation and activity which is fruitful for inhibiting bone resorption in bone losing disorders [116].

#### 4.3.5. Prostaglandins E2 Receptor Agonist and Antagonist

Prostaglandins (PG) E2 receptors are the most versatile prostanoids that are produced mainly in osteoblasts [117]. They have the ability to promote both bone formation and bone resorption. They contain four subtypes including EP1, EP2, EP3, and EP4, encoded by distinct genes and expressed differentially in tissues. It has been shown that among all four subtypes, PGE2-EP4 is the most active player in bone remodeling, and thus may act as a therapeutic target for different agonists and antagonist to treat bone disorders [118]. Recently, Caselli and colleagues reported CR6086 as a novel selective EP4 receptor antagonist that can be used as disease-modifying antirheumatic drugs (DMARDs) to treat rheumatoid arthritis [119].

#### 4.3.6. RANKL Inhibitors

RANKL is actually a transmembrane protein that belongs to tumor necrosis factor (TNF) cytokine family. It acts as a ligand for receptor RANK to induce the function, formation, and survival of osteoclasts [120]. Therefore, it is utilized as a target site for RANKL inhibitors to discourage bone resorption in disorders like osteoporosis. Denosumab, recently approved by the FDA is one of the key RANKL inhibitors for the treatment of bone-related disorders. Actually, denosumab is a human monoclonal antibody that has the ability to inhibit the interaction of RANKL with RANK and thus disturbs osteoclastogenesis and the bone-resorbing capacity of mature osteoclasts [121].

#### 4.3.7. Carbonic Anhydrases Inhibitors

Carbonic anhydrases constitute a superfamily of metalloenzymes that generate bicarbonate (HCO_3_^−^) and protons (H^+^) through reversible hydration of carbonic anhydride [122]. They are expressed in different tissues with different proportions and forms. They are required by osteoclasts for providing protons to promote bone resorption. Therefore, inhibition of carbonic anhydrases with specific inhibitors is useful for treating bone disorders. Recently, Abdoli and his colleagues reacted 4-sulfamoyl benzoic acid with amines and amino acids and got final compounds called benzamide-4-sulfonamides. After investigation they declared these sulfonamides as effective inhibitors against carbonic anhydrases [123].

## 5. Future Directions and Concluding Remarks

Usually, bone treatment with systemic administrations requires high doses and cause toxicities in non-skeletal tissues and organs. Therefore, various targeting delivery systems have been developed that target specifically bone or bone cells to minimize off-target effects and improve efficacy. But development of an absolute viral or non-viral carrier system that can deliver gene, siRNA, and drugs to specific sites is still in progress or under investigation. However, in an experiment siRNA was delivered to a mouse model against semaphorin 4D which resulted in increased number of active osteoblast cells and bone mass [124].

Although non-viral vectors have low transfection efficiency they are safer and easily producible as compared to non-viral vectors. Non-viral vectors like liposome and cationic polymers are surprisingly too efficient in carrying cargo towards different cell organelles. They have been proven as a successful delivery system for bone disorder treatments in many of the in vitro and some of the in vivo experiments. However, they can be designed and engineered in different ways to encapsulate various types of therapeutic inhibitors and carry them to specific target sites to cure severe bone disorders. Moreover, gene therapy can play a tremendous role in targeting various bone locations more specifically and efficiently than any other technique in the future.

## Figures and Tables

**Figure 1 ijms-20-00565-f001:**
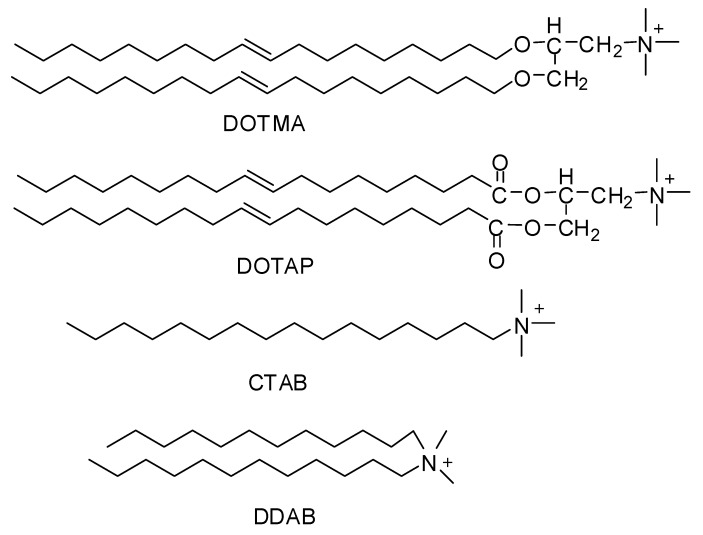
Chemical structures of cationic lipids.

**Figure 2 ijms-20-00565-f002:**
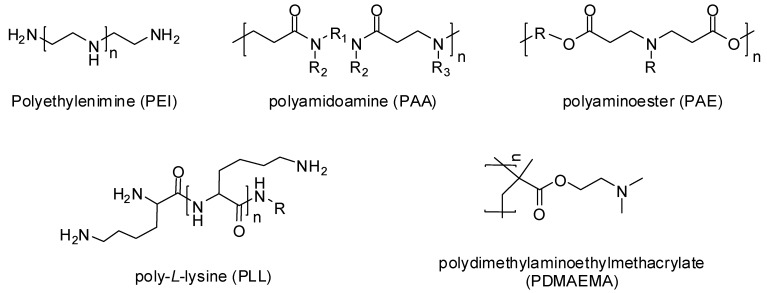
Chemical structures of cationic polymers.

**Figure 3 ijms-20-00565-f003:**
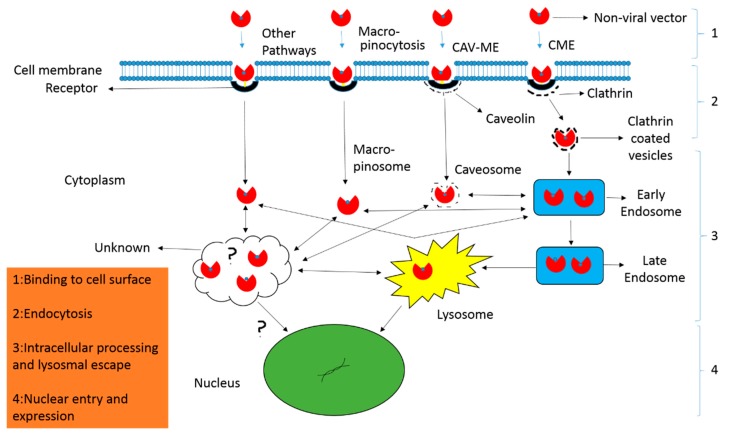
Mechanism of non-viral vector delivery. (**1**) The nanocarriers have to overcome extracellular barriers to reach and bind with cell surfaces either directly or following processing along filopodia. (**2**) After binding, the nanocarriers can enter the cell through various endocytic pathways including clathrin-dependent (CME) and clathrin-independent endocytosis. The latter refers to the caveolae-mediated endocytosis (CAV-ME), macropinocytosis and various other endocytic mechanisms. (**3**) The particles are processed inside the cell and release of contents occur from distinct endocytic compartments through various mechanisms like proton sponge effect (polyplexes), lipid mixing, and non-bilayer-induced membrane perturbation (lipoplexes). (**4**) In the final step, the contents are delivered to the nucleus for expression.

**Figure 4 ijms-20-00565-f004:**
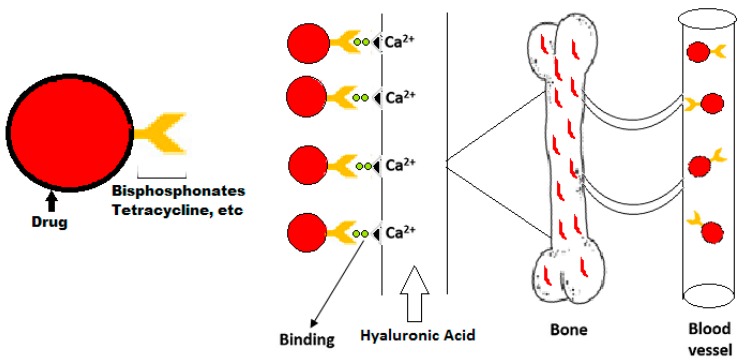
Schematic graph of delivery of bone specific drugs.

**Figure 5 ijms-20-00565-f005:**
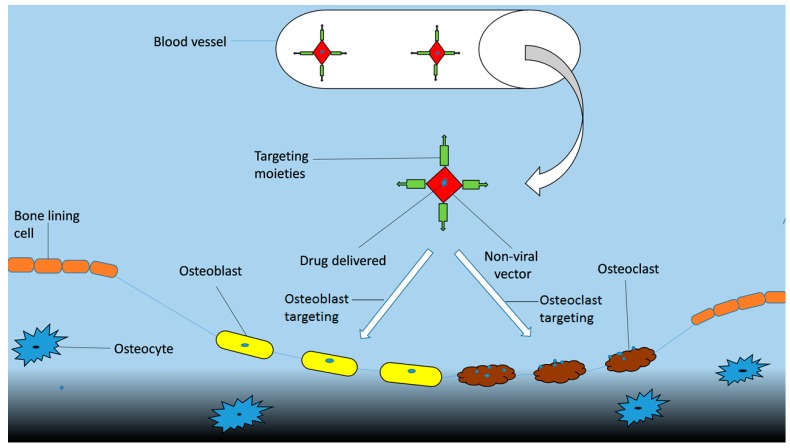
Scheme of delivery of bone cell specific drugs.

**Table 1 ijms-20-00565-t001:** Non-viral vectors as delivery systems against different bone disorders.

Non-Viral Vectors	Examples	Delivery to Bones and Disease Treatment	Limitations
Cationic lipids [10,11,12,13,14,15,16,17,18]	Chol-SSCOS/ES/DOX DOTMA, CTAB	Bone regeneration, Osteoporosis, Osteosarcoma	Hemotoxic, inflammatory response
Cationic polymers [19,20,21,22,23,24,25,26,27,28]	PEI, PAA, PAE, PLL, PDMAEMA	Mature osteoclasts suppression, Osteoporosis	Mostly nondegradable, Cytotoxic,
Cationic peptides [29,30,31]	SDSSD-PU, DSS6	Targeting osteoblasts, Osteoblasts related disorders	Circulation time
Calcium phosphates [32,33,34,35]	Calcium phosphate cements (CPCs)	Resorption surfaces, Osteoarthritis, Osteoporosis	Calcium toxicity, Osteosarcoma
Metal nanoparticles [36,37,38,39,40,41]	Au NPS, Fe_3_O_4,_ Fe_2_O_3_ NPS	Target BMCs and Suppress osteoclast formation, Osteoporosis	Non-degradable, potential toxicity

**Table 2 ijms-20-00565-t002:** Therapeutic agents targeting different molecules in bone cells.

Therapeutic Agents	Examples	Targeting Molecules	Bone Cells Expressing Targeting Molecules
Cathepsin K inhibitor [103,104,105,106]	MIV-711	Cathepsin K	Osteoclast
V-ATPase inhibitor [107,108]	Luteolin	V-ATPase	Osteoclast
α_v_β_3_ integrin receptor antagonist [109,110,111,112,113]	Dual-specific M-CSF (Macrophage colony-stimulating factor) mutants	α_v_β_3_ integrin receptor	Osteoclast
Src SH2 inhibitor [114,115,116]	AZD0530	Src SH2	Osteoclast
Prostaglandine receptor agonist and antagonist [117,118,119]	CR6086	Prostaglandine receptor	Osteoblast
RANKL inhibitor [120,121]	Denosumab	RANKL	Osteoclast
Carbonic anhydrase inhibitor [122,123]	Benzamide-4-sulfonamides	Carbonic anhydrase	Osteoblast

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
