# Peer review of "Non-Viral Delivery System and Targeted Bone Disease Therapy"

_ijms, 2019, doi:10.3390/ijms20030565_

Round 1
Reviewer 1 Report
Authors provide a nice review of some used non-viral delivery systems. I am not sure if the figures are original work of authors, if not, references should be added. I have a factual concern to the statement in lines 163-165. This is a generarly assumed non-accurate statement given by non-accurate interpretation of the original paper that studied PEI as transfection agent and now is appearing as a fact in many publications related to non-viral transfection systems. The situation is more complex and depends on the length of the DNA transfected and the used biological system in general. Also low Mw PEI can be very efficient for transfection. Please clarify and provide references for your statements. Please specify author contribution and funding.
Author Response
Dear reviewer,
I am highly grateful to your valuable comments. The answer to your question is as follows. I hope that my answer will satisfy your questions.
Question (1):
Answer: Dear reviewer I am highly grateful to your valuable comments. Actually I designed the figures myself so no need to give the references. Moreover, I revised 163-165 lines in the previous manuscript and studied it according to your suggestions. Yes, you were right therefore, I studied some other related articles and added accordingly in lines 177-186. We provided author contribution and funding.
Reviewer 2 Report
The have submitted a review dealing with 'Non-viral delivery system and targeted bone disease
therapy'. Basically, the review is worth and shows interesting aspects. However, I suggest to change the following before publication.
Minor points:
The authors write in the abstract '...several life threatening bone disorders like osteoporosis, osteoarthritis...' osteoporosis and osteoarthritis aren't life threatening perse - please change wording.
Format (font size) is different in some parts and should be changed, e.g.; page 5, lines 163 - 168 'Generally, cationic polymers with low molecular weights are preferred because they are tolerated nicely by the cells but have a drawback of lower transfection efficacy. Recently, Liu and his fellows solved this issue by synthesizing low molecular weight cationic polymers (LCPs) that was modified with coordinated zinc. They reported that Zn coordinated ligand bind with high affinity to phosphate group in DNA that help the formulated polyplex in endosomal escape and thus increasing transfection efficacy..'
Major points:
My main concern with this work is the the dysbalance between textbook knowledge and bone relation. This should be changed and linked with the bone, respectively shown in tables.
The chapters '2. Non-viral delivery system' and '3. Mechanism of non-viral delivery system' are more or less text book knowledge and don't show the link to bone. The authors shall add a Table describing the technology and linking it to a bone application with corresponding reference/s.
It is unclear why the authors have discussed 'Bisphosphonates'. They have nothing to do non-viral delivery?
In the actual Table 1, the authors should add references underlining the importance of the summarized data.
Author Response
Reply to reviewer 2
Dear reviewer,
I am highly grateful to your valuable comments. The answer to your question is as follows. I hope that my answer will satisfy your questions.
Question (1):
Answer: yes you are right that osteoporosis and osteoarthritis are not life threatening disorders. Therefore I changed the word “life threatening’’ with “life suffering” as shown in line (17) of the manuscript.
Question (2):
Answer: The format of the line (163-168) in previous manuscript has been changed and made similar with that of the overall format as shown in (177-186).
Question (3):
Answer: yes you are right that there was somewhat dysbalance in between the title and description in some parts of the article. However, by studying some extra articles, I added and linked each vector with delivery to bone according to the available literature as shown in lines (86-88), (98-99), (131-139), (144-145), (148-149), (186-191), (206-212), (244-252). I removed those references which were general and not linking to bone. I added a table1containng references that clearly shows the use of different vectors for delivery to bone to treat various bone disorders. Moreover, I tried to linked mechanism of nonviral vector delivery there was necessary. However, the basic mechanism for all the cells whether bone cells or other cells is similar in literature with only slight differences.
Question (4):
Answer: Although bisphosphonates are not nonviral vectors but act as important bone targeting moiety that can make conjugates with different nonviral vectors to guide them to reach specific bone sites to deliver drug. Actually they are negatively charged due to their phosphate groups and have a strong affinity to positively charged calcium of hydroxyapatite crystals of bone. Therefore I discussed it in bone targeting system part of the review.
Question (5):
Answer
Answer: In the actual table1, I have added all the required references which have been shown in the manuscript.